# Fair Classification with Instance-dependent Label Noise

**Songhua Wu**                                                    SONGHUA.WU@SYDNEY.EDU.AU
*Trustworthy Machine Learning Lab*
*The University of Sydney*
*Sydney, Australia*

**Mingming Gong**                                               MINGMING.GONG@UNIMELB.EDU.AU
*School of Mathematics and Statistics*
*The University of Melbourne*
*Melbourne, Australia*

**Bo Han**                                                       BHANML@COMP.HKBU.EDU.HK
*Department of Computer Science*
*Hong Kong Baptist University*
*Hong Kong, China*

**Yang Liu**                                                     YANGLIU@UCSC.EDU
*Computer Science and Engineering*
*University of California, Santa Cruz*
*Santa Cruz, United States*

**Tongliang Liu**                                                TONGLIANG.LIU@SYDNEY.EDU.AU
*Trustworthy Machine Learning Lab*
*The University of Sydney*
*Sydney, Australia*

**Editors:** Bernhard Schölkopf, Caroline Uhler and Kun Zhang

## Abstract

With the widespread use of machine learning systems in our daily lives, it is important to consider fairness as a basic requirement when designing these systems, especially when the systems make life-changing decisions, e.g., *COMPAS* algorithm helps judges decide whether to release an offender. For another thing, due to the cheap but imperfect data collection methods, such as crowdsourcing and web crawling, label noise is ubiquitous, which unfortunately makes fairness-aware algorithms even more prejudiced than fairness-unaware ones, and thereby harmful. To tackle these problems, we provide general frameworks for learning fair classifiers with *instance-dependent label noise*. For statistical fairness notions, we rewrite the classification risk and the fairness metric in terms of noisy data and thereby build robust classifiers. For the causality-based fairness notion, we exploit the internal causal structure of data to model the label noise and *counterfactual fairness* simultaneously. Experimental results demonstrate the effectiveness of the proposed methods on real-world datasets with controllable synthetic label noise.

**Keywords:** causal graph; counterfactual fairness; instance-dependent label noise.

## 1. Introduction

Machine learning systems have been widely adopted in our daily life. The overwhelming advantages for these systems are that they never get tired, and they approach (and sometimes surpass) human-level benchmarks on a wide array of tasks (Danziger et al., 2011; Silver et al., 2016). Thereby, they are entrusted with important tasks, i.e., making high-stakes decisions in loan applications (Mukerjee et al., 2002), dating and hiring (Bogen and Rieke, 2018; Cohen et al., 2019), and even parole (Dressel and Farid, 2018). Nevertheless, machine learning algorithms are very sensitive to biases which render their decisions *unfair*[1] (Mehrabi et al., 2021; Angwin et al., 2016; O'neil, 2016). One canonical example is a decision support tool used by U.S. courts to assess the likelihood of a defendant becoming a recidivist, called COMPAS (Dressel and Farid, 2018). A bias against African-Americans was found with this software in an analysis performed by the news organization ProPublica: COMPAS is more likely to assign a higher risk score to African-American offenders than to Caucasians with the same profile.

To mitigate the bias in machine learning algorithms, plenty of methods, that can be roughly divided into two broad groups, have been proposed. The first group of methods focuses on the statistical fairness notions, which discover the discrepancy of statistical metrics between individuals or sub-populations, e.g., statistical parity (Dwork et al., 2012), equalized odds (Hardt et al., 2016), and predictive parity (Chouldechova, 2017). This group of methods only considers the correlation but ignores causal effect relations within the data, which can hardly assess the fairness sufficiently (Huan et al., 2020). The second group of methods focuses on the causality-based fairness notions, which additionally employs *causal graphs* to take knowledge about the structure of real-world datasets into consideration (Makhlouf et al., 2020), e.g., fair on average causal effect (Khademi et al., 2019), counterfactual fairness (Kusner et al., 2017), and counterfactual error rates (Zhang and Bareinboim, 2018).

However, the above methods are based on a strong assumption that labels are entirely accurate, which is hard to achieve due to the way labels were generated, e.g., the ImageNet-scale dataset was necessarily annotated by distributed workers in Amazon Mechanical Turk[2] (Han et al., 2020b). Northcutt et al. (2021) identified an average of 3.4% label noise in the test sets of 10 of the most commonly-used computer vision, natural language, and audio datasets. It is well-known that label noise degenerates the performance of deep networks, because deep networks easily overfit label noise (Zhang et al., 2017; Han et al., 2020a; Wu et al., 2021; Bai et al., 2021; Xia et al., 2021). Lamy et al. (2019); Fogliato et al. (2020); Wang et al. (2021); Liu and Wang (2021) designed experiments to demonstrate that naively enforcing parity constraints on the noisy labels harms the accuracy of the classifier for the groups that are not affected by label noise. To make things worse, label noise also degenerates the fairness metrics and could make some fairness-aware algorithms even more prejudiced than fairness-unaware ones.

To see this, first, we add two types of label noise, i.e., class-dependent label noise (CDLN) and instance-dependent label noise (IDLN), onto a benchmark dataset $ADULT$[3] (Dua and Graff, 2017). For class-dependent label noise, given clean label $Y$, the noisy label $\tilde{Y}$ is conditionally independent of the instance $X$, i.e., $P(\tilde{Y} \mid Y, X) = P(\tilde{Y} \mid Y)$. Instance-dependent label noise is more complex and can capture the true structure of real-world datasets better. The noise rates are set to 0.3 and

---

1. Fairness is the absence of any prejudice or favoritism toward an individual or a group based on their inherent or acquired characteristics (Saxena et al., 2019).

2. https://www.mturk.com/

3. The ADULT dataset is from UCI ML Repository with gender as the sensitive attribute

Table 1: Means and Stds of classification accuracy and fairness score ($p$ value. The higher the value, the better the fairness.) on ADULT dataset with two kinds of label noise over 5 trials. UC denotes the method which optimizes the training loss unconstrainedly.

|  | CDLN-0.3 | | CDLN-0.4 | | IDLN-0.3 | | IDLN-0.4 | |
|---|---|---|---|---|---|---|---|---|
|  | Accuracy | Fairness | Accuracy | Fairness | Accuracy | Fairness | Accuracy | Fairness |
| $p$-Fair | 80.46±0.35 | 28.37±4.28 | 79.90±0.33 | 31.44±7.42 | 80.35±0.26 | 24.18±6.14 | 79.86±0.29 | 25.07±9.87 |
| UC | 83.47±0.30 | 28.89±4.14 | 82.76±0.42 | 32.03±6.20 | 83.38±0.24 | 25.93±5.46 | 82.84±0.45 | 26.37±8.23 |

0.4:

$$P(\tilde{Y} = -1 \mid Y = 1, X) = P(\tilde{Y} = 1 \mid Y = -1, X) = 0.3\ (0.4). \tag{1}$$

Then we implement the algorithm ($p$-Fair) in Zafar et al. (2017) to learn a fair classifier. Zafar et al. (2017) considered two distinct notions: disparate treatment and disparate impact (Barocas and Selbst, 2016), and employed $p\%$-rule:

$$\min\left(\frac{P\left(\hat{Y} = 1 \mid A = 1\right)}{P\left(\hat{Y} = 1 \mid A = 0\right)}, \frac{P\left(\hat{Y} = 1 \mid A = 0\right)}{P\left(\hat{Y} = 1 \mid A = 1\right)}\right) \geq \frac{p}{100}, \tag{2}$$

as a constraint in the objective function, where $\hat{Y}$ is the predicted label and $A$ is the protected attribute. As shown in Table 1, the fairness-aware method ($p$-Fair) gives more unfair and misleading decisions than the vanilla unconstrained method (UC) under the influence of both kinds of label noise. At the same noise rate, IDLN is more harmful and thus more challenging.

For fairness-aware algorithms employing causality-based fairness notions, the fairness metrics could be robust to label noise to some extent. For example, counterfactual fairness (Kusner et al., 2017) requires that changing the value of protected attribute $A$, while holding things that are not causally dependent on $A$ constant, will not change the distribution of the predicted label. One straightforward strategy to achieve counterfactual fairness is to build a classifier only consisting of the non-descendants of $A$. From Figure 2, we can see that the label noise does not change the internal causal structure of instances. The original non-descendant $Z$ is still the non-descendant of $A$,

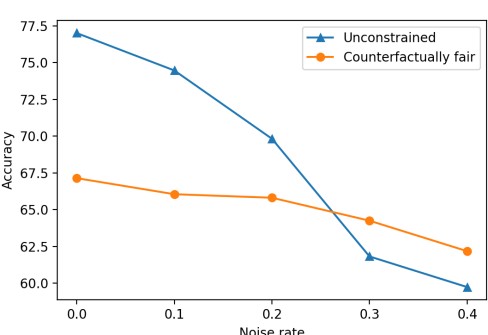

Figure 1: Means of classification accuracy on ADULT dataset over 5 trials.

which means the classifier built only with $Z$ is robust to label noise with respect to the counterfactual fairness. Although the fairness is maintained, the decline in accuracy is unavoidable. Figure 1 shows that the classification accuracy of the classifier only using non-descendants $Z$ decreases as the noise rate increases. Especially when the data are clean, the gap between the counterfactually

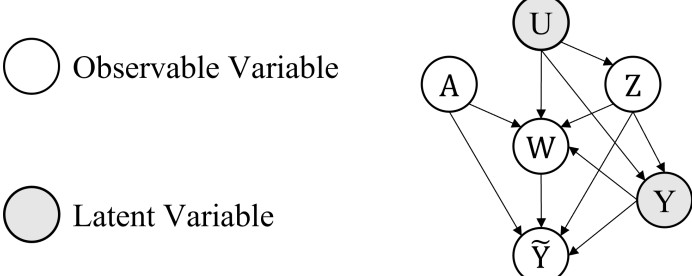

Figure 2: Postulated causal graph. Label noise does not change the internal structure of $(A, W, Z)$.

fair classifier and the unconstrained classifier is huge, indicating there is a huge information loss of the counterfactually fair classifier.

In this paper, we provide general frameworks for learning fair classifiers with instance-dependent label noise. For statistical fairness notions, we rewrite the classification risk and the fairness metric in terms of noisy data and thereby build robust classifiers. For the causality-based fairness notion, we exploit the internal causal structure of data to model the label noise and counterfactual fairness (Kusner et al., 2017) simultaneously. Specifically, we postulate a general causal graph as shown in Figure[4] 2 and employ the variational autoencoder (VAE) framework (Kingma and Welling, 2013) to make full use of the causal graph which can infer latent variables $U$ and $Y$ by maximizing the joint likelihood of observable variables. In this way, our method also compensates for the information loss, because $W$ contains information from its parents $A$ and $U$, and we extract the $U$-part information in $W$ by reconstructing $W$ with $U$.

The rest of this paper is organized as follows: In Section 2, we formulate the problem of fair classification with label noise. In Section 3 and 4, we provide general frameworks for learning fair classifiers with instance-dependent label noises for statistical fairness notions and the counterfactual fairness notion, respectively. Experimental results are discussed in Section 5. We conclude our paper in Section 6.

## 2. Preliminaries

We consider the binary fair classification problem. Let $\mathcal{D}$ be the distribution of a pair of random variables $(X, Y) \in \mathcal{X} \times \{-1, 1\}$, where $\mathcal{X} \subset \mathbb{R}^d$ and $d$ represents the feature dimension. $X$ can be denoted in detail as a triple $(A, Z, W)$, where $A$ is a protected attribute; $Z$ is a non-descendant variable of $A$, denoting some root-level attributes; $W$ is the low-level attributes. In real-world datasets, the clean label cannot be observed. Instead, we can only observe the noisy label $\tilde{Y}$. In this case, we have a sample $\{(a_1, z_1, w_1, \tilde{y}_1), \ldots, (a_n, z_n, w_n, \tilde{y}_n)\}$ drawn from a noisy distribution $\mathcal{D}_\rho$ of the random variables $(A, Z, W, \tilde{Y})$ as shown in Figure 2. We aim to learn a robust and fair classifier that could assign clean labels to test data by exploiting the sample with noisy labels.

---

4. We will take benchmark dataset *ADULT* (Dua and Graff, 2017) as an example to demonstrate how this causal graph interprets the data in Section 4.2.

## 2.1. Label Noise

The label noise structure is usually formulated by a $C \times C$ transition matrix, where $C$ is the number of classes. The $ij$-th element of a transition matrix is $T_{ij}(x) = P(\tilde{Y} = j \mid Y = i, X = x)$, which represents the probability that the instance $x$ with the clean label $Y = i$ actually has a noisy label $\tilde{Y} = j$. It can establish the connection between noisy posterior and clean posterior, i.e., $P(\tilde{Y} \mid X) = T^\top P(Y \mid X)$. Utilizing a transition matrix, consistent algorithms can be built (Natarajan et al., 2013; Scott, 2015; Liu and Tao, 2016; Patrini et al., 2017; Li et al., 2021). However, without assumptions, the transition matrix is not identifiable (Xia et al., 2019). The most commonly used assumption is the class-dependent noise assumption: given the clean label $Y$, the noisy label $\tilde{Y}$ is conditionally independent of instance $X$, i.e., $P(\tilde{Y} = j \mid Y = i, X = x) = P(\tilde{Y} = j \mid Y = i)$. Under this assumption, one can use anchor points (Liu and Tao, 2016) to estimate the transition matrix. Nevertheless, the real-world noise is barely class-dependent, which narrows down the application scenarios of this simplified label noise model. In this paper, we study the practical instance-dependent label noise, and we make use of the causal graph to help to identify the transition relationship $P(\tilde{Y} = j \mid Y = i, X = x)$.

## 2.2. Causal Graph and Structural Causal Model

We follow (Pearl et al., 2000) to use the Directed Acyclic Graph (DAG) with arrows pointing from the parent (direct cause) node to the child (direct effect) node as a formalism to represent causal relationships. Based on the DAG, we use structural causal models (SCMs) to represent the causal mechanism underlying the data distribution: variables can be expressed by a function of their parents with exogenous noise. For Figure 2, the corresponding structural causal model can be written as

$$Z = f(U, \varepsilon_Z), \quad Y = f(U, Z, \varepsilon_Y), \quad W = f(U, A, Z, Y, \varepsilon_W), \quad \tilde{Y} = f(A, Z, Y, W, \varepsilon_{\tilde{Y}}). \quad (3)$$

Each equation captures a conditional distribution of the term on the left side, conditioned on terms on the right side (excluding the exogenous variable). Note that the last equation is exactly representing the transition relationship $P(\tilde{Y}|A, Z, Y, W)$ we want to identify.

## 3. Statistically Fair Classification with Instance-dependent Label Noise

In this section, we demonstrate how to rewrite the classification risk and the fairness metric in terms of noisy data and thereby build robust classifiers for statistical fairness notions.

For almost all the statistically fair classification problem, they can be formulated by a constrained optimization problem. Generally, we minimize the classification error $L(\cdot)$ on training data subject to a specific statistical fairness constraint $\text{Fair}(\cdot)$:

$$\begin{aligned} \text{minimize} \quad & \sum_{n=1}^{N} L(f(x_n), y_n) \\ \text{subject to} \quad & \text{Fair}(X, Y, f) = 0. \end{aligned} \quad (4)$$

The clean optimization problem can be statistically linked to the noisy optimization problem with the transition relationship. Next, we propose two general methods and for illustration, we specialize them for two representative fairness notions: equalized odds and $p$-Fair, respectively. Methods designed for them can be easily extended to equal opportunity and demographic parity (Hardt

et al., 2016; Dwork et al., 2012; Verma and Rubin, 2018).

**Equalized Odds** (Hardt et al., 2016). For the classification error part, we show how to use importance reweighting technique (Bruzzone and Marconcini, 2009; Liu and Tao, 2016) to consistently estimate it:

$$
\begin{aligned}
&\mathbb{E}_{(X,Y)\sim\mathcal{D}}[L(f(X),Y)] \\
&= \int P_{\mathcal{D}}(X,Y)L(f(X),Y)\mathrm{d}X\,\mathrm{d}Y \\
&= \int P_{\mathcal{D}_\rho}(X,Y)\frac{P_{\mathcal{D}}(X,Y)}{P_{\mathcal{D}_\rho}(X,Y)}L(f(X),Y)\mathrm{d}X\,\mathrm{d}Y \\
&= \mathbb{E}_{(X,Y)\sim\mathcal{D}_\rho}\left[\frac{P_{\mathcal{D}}(X,Y)}{P_{\mathcal{D}_\rho}(X,Y)}L(f(X),Y)\right] \\
&= \mathbb{E}_{(X,Y)\sim\mathcal{D}_\rho}[\beta(X,Y)L(f(X),Y)],
\end{aligned}
\tag{5}
$$

where $\beta(x,y) = \frac{P_{\mathcal{D}}(X=x,Y=y)}{P_{\mathcal{D}_\rho}(X=x,Y=y)}$. To calculate $\beta(x,y)$, we only need noisy data and the noise rate. Let

$$
T_{-1}(x) = P(\tilde{Y} = +1 \mid Y = -1, X = x), \ \ T_{+1}(x) = P(\tilde{Y} = -1 \mid Y = +1, X = x),
$$

then we have

$$
P(\tilde{Y} = y \mid X = x) = (1 - T_{-1}(x) - T_{+1}(x))\,P(Y = y \mid X = x) + T_{-y}(x)
$$

and

$$
\beta(x,y) = \frac{P(\tilde{Y} = y \mid X = x) - T_{-y}(x)}{(1 - T_{-1}(x) - T_{+1}(x))\,P(\tilde{Y} = y \mid X = x)}.
\tag{6}
$$

For the equalized odds constraint part, the original one is $|\gamma_0\left(\hat{Y}\right) - \gamma_1\left(\hat{Y}\right)| = 0$, where $\gamma_a(\hat{Y}) \triangleq \{P(\hat{Y} = 1 \mid A = a, Y = 1), P(\hat{Y} = 1 \mid A = a, Y = 0)\}$. Now we rewrite the first term of $\gamma_0\left(\hat{Y}\right)$:

$$
\begin{aligned}
&P(\hat{Y} = 1 \mid A = a, Y = 1) \\
&= \frac{P(\hat{Y} = 1, A = a, Y = 1)}{P(A = a, Y = 1)} \\
&= \frac{P(\hat{Y} = 1, A = a, Y = 1)P(A = a, \hat{Y} = 1)}{P(A = a, Y = 1)P(A = a, \hat{Y} = 1)} \\
&= \frac{P(Y = 1 \mid A = a, \hat{Y} = 1)P(A = a, \hat{Y} = 1)}{P(A = a, Y = 1)} \\
&= \frac{\left(P(\tilde{Y} = 1 \mid A = a, \hat{Y} = 1) - T_{-1}(a)\right)P(A = a, \hat{Y} = 1)\,(1 - T_{-1}(a) - T_{+1}(a))}{(1 - T_{-1}(a) - T_{+1}(a))\left(P(\tilde{Y} = 1, A = a) - T_{-1}(a)\right)} \\
&= \frac{\left(P(\tilde{Y} = 1 \mid A = a, \hat{Y} = 1) - T_{-1}(a)\right)P(A = a, \hat{Y} = 1)}{P(\tilde{Y} = 1, A = a) - T_{-1}(a)},
\end{aligned}
\tag{7}
$$

where all variables are accessible, either observable or learnable, and

$$T_{-1}(a) = P(\tilde{Y} = +1 \mid Y = -1, A = a), \ \ T_{+1}(a) = P(\tilde{Y} = -1 \mid Y = +1, A = a).$$

Note that this group transition relation $T_y(a)$ can be derived from the individual one $T_y(x)$. The detailed derivation process is provided in Appendix A.

The rest three terms can be rewritten in a similar way. At this point, we can use noisy data to learn a robust classifier with equalized odds.

$p$-**Fair** (Zafar et al., 2017). For the classification error part, we show how to employ a transition matrix to learn a consistent classifier (Patrini et al., 2017). Let $f(X)$ output the posterior of $Y \in \{-1, 1\}$, i.e., $f(X) = P(Y \mid X)$, then $P(\tilde{Y} \mid X) = T^\top(X)f(X)$. Therefore, by minimizing $L(T^\top(X)f(X), \tilde{Y})$, the learned $f$ is consistent with the one learned on clean data.

For the $p$-Fair fairness constraint, the original one is Demographic Parity $|P(\hat{Y}|A = 0) - P(\hat{Y}|A = 1)| = 0$. In practice, they use a soft one $|\frac{1}{N} \sum_{i=1}^{N} (a_i - \bar{a}) f(x)| \leq c$, where $c$ is a threshold. Note that the consistent classifier $f$ is for clean data, which means we can directly substitute it to the constraint. We name this modified method *Robust-$p$-Fair* (R-$p$-Fair).

In practical implementation, we employ the Lagrange multipliers method (Bertsekas, 2014) to transfer a constraint to a regularization term. If the instance-dependent transition matrix is not given, we can approximate it for one instance by a combination of the transition matrices for the parts of the instance. Estimating the transition matrix will be much easier if a small clean set is given (Xia et al., 2019, 2020).

## 4. Counterfactually Fair Classification with Instance-dependent Label Noise

In this section, we consider the causality-based fairness notion. We elaborate on how to make full use of the causal graph to design a robust and counterfactually fair classifier. Then we showcase how to implement the algorithm in practice.

### 4.1. Counterfactual Fairness

Intuitively, counterfactual fairness requires that changing $A$, while holding things that are not causally dependent on $A$ constant, will not change the distribution of the predictor $h$:

**Definition 1** *(Kusner et al., 2017) Predictor $h$ is counterfactually fair if under any context $X = x$ and $A = a$,*

$$P\left(h_{A \leftarrow a}(U) = y \mid X = x, A = a\right) = P\left(h_{A \leftarrow a'}(U) = y \mid X = x, A = a\right)$$

*for all $y$ and for any value $a$ $a'$ attainable by $A$.*

One straightforward strategy to achieve counterfactual fairness is the following:

**Lemma 2** *(Kusner et al., 2017) Let $\mathcal{G}$ be the causal graph of the given model. Then classifier will be counterfactually fair if it is a function of the non-descendants of $A$.*

A major concern of this strategy is that it totally discards $W$ and loses much information. $W$ inherently contains information from its parents $A$ and $U$, and we extract the $U$-part information in $W$ by reconstructing $W$ with $U$.

## 4.2. How the Causal Graph Interprets the Data

Here we take the benchmark dataset *ADULT* (Dua and Graff, 2017) as an example to demonstrate how this causal graph (Figure 2) interprets the data:

- $A$ represents the protected attribute 'Gender'.

- $Z$ represents the other root-level attributes, i.e., 'Age', 'Race', and 'Native country', which are not affected by the protected attribute $A$.

- $W$ represents the low-level attributes, e.g., 'Workclass', 'Capital-gain', and 'Marital-status', which are caused by the background variable $U$ and root-level attributes $A$ and $Z$: $U \rightarrow W \leftarrow (A, Z)$.

- $U$ is a latent variable and can be seen as 'Background' of people, which causes those non-protected attributes, making $U$ a confounder of $W$ and $Z$: $W \leftarrow U \rightarrow Z$.

- $Y$ represents the clean but latent label, the annual income, which is influenced by the background and root-level attributes of people: $U, Z \rightarrow Y$. Note that, to fulfill counterfactual fairness, we intentionally block the path from $A$ to $Y$. Meanwhile, annual income (not the salary of a job) acts as a cause of the low-level attributes: $Y \rightarrow W$. For example, people with lower annual income are less willing to do 'Without-pay' work, which is one kind of 'Workclass'. For another example, people with higher annual income pay more attention to investment and wealth management and thereby have a larger 'Capital-gain'. Besides, annual income can obviously affect 'Marital-status'.

- $\tilde{Y}$ represents the noisy label, which is a common child of the observable variables and clean label: $(A, Z, W, Y) \rightarrow \tilde{Y}$.

Based on this causal graph, we can only feed $U$ and $Z$ to the classifier $f$ to infer the clean label $Y$, which, according to the Lemma 2, makes $f$ counterfactually fair. Specifically, we employ the variational autoencoder (VAE) framework (Kingma and Welling, 2013) to make full use of the causal graph which can infer latent variables $U$ and $Y$ by maximizing the joint likelihood of observable variables. Moreover, exploiting the causal graph contributes to the identifiability of the transition relationship between clean and noisy labels (Yao et al., 2021).

## 4.3. VAE based Causal Inference

The joint distribution $p(U, A, Z, W, \tilde{Y}, Y)$ specified by the causal graph in Figure 2 and the structural causal model Eq. (3) can be factorized as follows:

$$
\begin{aligned}
&p(U, A, Z, W, \tilde{Y}, Y) \\
&= p(A)p(U)p(Z \mid U, A)p(Y \mid U, A, Z)p(W \mid U, A, Z, Y)p(\tilde{Y} \mid U, A, Z, Y, W) \quad (8) \\
&= p(A)p(U)p(Z \mid U)p(Y \mid U, Z)p(W \mid U, A, Z, Y)p(\tilde{Y} \mid A, Z, Y, W).
\end{aligned}
$$

Note that although $A$ is involved in the reconstruction of $W$ and $\tilde{Y}$, it does not causally affect how $U$ and $Z$ infer $Y$. Namely, the counterfactual fairness still holds.

In the encoding phase, we infer the latent variable $U$ and $Y$ from observable variables $Z$. Without loss of generality, we choose prior $p(U)$ to be simple, i.e., Gaussian. We use an encoder with a

learnable parameter $\phi$ to model the distribution $p(U, Y \mid A, Z, W, \tilde{Y})$. Since $A$ and its descendant $W$ are not allowed to build the classifier, and given its all parents $U$ and $Z$, $Y$ is independent on $(A, W, \tilde{Y})$, the encoder can be simplified as:

$$
\begin{aligned}
q_\phi(U, Y \mid A, Z, W, \tilde{Y}) &= q_{\phi_1}(U \mid A, Z, W, \tilde{Y}) q_{\phi_2}(Y \mid U, A, Z, W, \tilde{Y}) \\
&= q_{\phi_1}(U \mid Z) q_{\phi_2}(Y \mid U, Z),
\end{aligned}
\tag{9}
$$

where $q_{\phi_2}(Y \mid U, Z)$ can be employed as a counterfactually fair classifier $f$.

In the decoding phase, given that $p(U)$ is Gaussian, we need four decoders corresponding to the rest four terms on the right side of Eq. (8), as:

$$
p_\theta(U, A, Z, W, \tilde{Y}, Y) = p(A)p(U)p_{\theta_1}(Z \mid U)p_{\theta_2}(Y \mid U, Z)p_{\theta_3}(W \mid U, A, Z, Y)p_{\theta_4}(\tilde{Y} \mid A, Z, Y, W).
\tag{10}
$$

We denote $\Theta = \{\phi_1, \phi_2, \theta_1, \theta_2, \theta_3, \theta_4\}$ the parameter set of this VAE network. In the evaluation phase, we first sample $U$ from $q_{\phi_1}(U \mid Z)$ and then use $(U, Z)$ to infer $Y$ with $q_{\phi_2}(Y \mid U, Z)$. Note that $\phi_2$ and $\theta_2$ are the same, which both model the generation process of $Y$. It is because $Y$ is a latent intermediate variable such that modeling $Y$ can be treated as either encoding or decoding. Hereinafter we refer to them collectively using classifier $f$.

Then, because the data likelihood $p_\Theta(A, Z, W, \tilde{Y})$ is intractable, instead of maximizing the data likelihood, we learn $\Theta$ by minimizing the negative evidence lower bound (ELBO) (Kingma and Welling, 2013). ELBO is a lower bound of the likelihood, which is preferred for optimization because it can be calculated efficiently.

Starting with maximizing the data likelihood $p_\Theta(A, Z, W, \tilde{Y})$, we can derive the negative ELBO as follows (the detailed derivation process is provided in Appendix B):

$$
-\text{ELBO} \triangleq -\mathrm{E}_{(u,y) \sim q_\phi(u,y|z)} \left[\log p_{\theta_1}(z \mid u)\right] - \mathrm{E}_{(u,y) \sim q_\phi(u,y|z)} \left[\log p_{\theta_3}(w \mid u, a, z, y)\right]
\tag{11}
$$

$$
- \mathrm{E}_{(u,y) \sim q_\phi(u,y|z)} \left[\log p_{\theta_4}(\tilde{y} \mid a, z, y, w)\right] + \mathrm{D}_{\text{KL}}(q_{\phi_1}(u \mid z) \| p(u)),
\tag{12}
$$

where $\mathrm{D}_{\text{KL}}$ is the Kullback–Leibler divergence function. Although the above ELBO does not explicitly involve the counterfactually fair classifier $f$, the prediction $Y$ plays an important role in the second and third terms of ELBO, which pushes $f$ to be optimized.

So far, the classifier outputs a counterfactually fair prediction $Y$, which can be treated as cluster numbers but not clean class labels. Since $Y$ is a latent intermediate variable, the map between the value of $Y$ ($+1$ or $-1$) to the semantic class (*positive* or *negative*) is lost. To map $Y$ to semantic clean labels, noisy labels $\tilde{Y}$ are the only thing we have that could help. In case $f$ is severely misled by $\tilde{Y}$, we introduce a data augmentation technique *Mixup* (Zhang et al., 2018), which generates a weighted combination of random instance pairs from the training data:

$$
\hat{x} = \lambda x_i + (1 - \lambda)x_j,
\tag{13}
$$

$$
\hat{y} = \lambda y_i + (1 - \lambda)y_j,
\tag{14}
$$

where weights $\lambda$ are independently sampled from a Beta distribution for each augmented example. Mixup prevents $f$ from overfitting noisy labels in two aspects. First, it increases the complexity of the training data, which makes it difficult for a network to learn. Second, by combining different features (labels) with one another, a network does not get overconfident about the relationship between the features and their labels.

---

**Algorithm 1** Robust Counterfactually Fair Classification (RCFC).

---

**Input:** A training sample of observable variables $(A, Z, W, \tilde{Y})$.

1: Encode $U$:
2:      $\mu, \sigma = q_{\phi_1}(Z)$                                      ▷ reparameterization trick
3:      $U = \mu + \sigma\epsilon$               ▷ where $\epsilon$ is an auxiliary noise variable $\epsilon \sim \mathcal{N}(0, 1)$
4: Encode $Y$:
5:      $Y = q_{\phi_2}(U, Z)$
6: Decode (Reconstruct) $Z, W, \tilde{Y}$:
7:      $\hat{Z} = p_{\theta_1}(U)$                          ▷ where $\hat{Z}$ is the predicted value of $Z$.
8:      $\hat{W} = p_{\theta_3}(U, A, Z, Y)$            ▷ where $\hat{W}$ is the predicted value of $W$.
9:      $\tilde{Y}^\diamond = p_{\theta_4}(A, Z, Y, W)$         ▷ where $\tilde{Y}^\diamond$ is the predicted value of $\tilde{Y}$.
10: Update the parameter set $\Theta$ by minimizing $-\text{ELBO}$ and the Mixup loss.
11: **Output:** Encoder $q_{\phi_1}(Z)$; Classifier $f$ (Encoder $q_{\phi_2}(U, Z)$).

---

### 4.4. Practical Implementation

The proposed algorithm is summarized in Algorithm 1.

For the negative ELBO part, the first three terms are exactly reconstruction errors (Kingma and Welling, 2013). Therefore, in practice, we use mean squared error to measure the reconstruction errors for $(\hat{Z}, \hat{W})$ with respect to $(Z, W)$, and we use cross-entropy loss to measure the reconstruction errors for $\hat{\tilde{Y}}$ with respect to $\tilde{Y}$. As for the last $\text{D}_{\text{KL}}$ term, first we use the reparameterization trick (Kingma and Welling, 2013) to sample $U$ once from $q_{\phi_1}(u \mid z)$, and $\mu, \sigma$ are continuous variables with gradients. Note that $U$ can also be the average value of several sampling results to decrease the variance. Then, we calculate $\text{D}_{\text{KL}}$ term with the closed-form solution provided by Kingma and Welling (2013):

$$\text{D}_{\text{KL}}(q_{\phi_1}(u \mid z) \| p(u)) = -\frac{1}{2} \sum_{j=1}^{J} \left( 1 + \log\left(\left(\sigma_j^{(i)}\right)^2\right) - \left(\mu_j^{(i)}\right)^2 - \left(\sigma_j^{(i)}\right)^2 \right), \quad (15)$$

where $J$ is the dimension of $U$.

For the mixup loss part, we first concatenate $U$ and $Z$ as input $I$, and then apply the mixup technique to classifier $f$ with pairs $(I, \tilde{Y})$. Here, we use cross-entropy loss.

### 5. Experiments

In this section, we examine how the proposed methods learn a robust fair classifier against instance-dependent label noise.

**Dataset.** We employ two widely used benchmark datasets:

- *ADULT* (Dua and Graff, 2017). The prediction task is to determine whether a person makes over \$50K a year, with gender as the protected attribute. The detailed information for this dataset and how it complies with the causal graph have been elaborated in Section 1.

- *BANK* (Dua and Graff, 2017). The prediction task is to determine whether a client subscribes to a term deposit, with gender as the protected attribute. We select personal attributes except

Table 2: Means and Standard deviations of classification accuracy and fairness score ($p$ value) on ADULT dataset over 5 trials.

| ADULT | IDLN-0.1 | | IDLN-0.2 | | IDLN-0.3 | | IDLN-0.4 | |
|---|---|---|---|---|---|---|---|---|
| | Accuracy | Fairness | Accuracy | Fairness | Accuracy | Fairness | Accuracy | Fairness |
| SSL | 71.63±5.16 | 14.22±5.48 | 63.68±5.02 | 21.24±6.87 | 58.80±4.17 | 25.90±5.09 | 51.20±8.13 | 34.45±6.05 |
| $p$-Fair | 69.46±6.83 | 30.08±7.19 | 61.89±4.96 | 32.85±5.92 | 58.50±4.42 | 35.10±5.53 | 49.85±7.83 | 40.43±4.97 |
| R-$p$-Fair | 69.97±7.18 | 41.78±1.02 | 63.27±3.94 | 38.35±3.69 | 59.46±4.67 | 39.74±5.08 | 51.42±8.78 | 41.02±4.47 |

gender and the credit history attributes as the other root-level attribute $Z$. Those loan-relevant and property-relevant attributes are selected as the low-level attributes $W$. We drop social and economic context attributes because they are irrelevant.

For all datasets, of which 10% are split as test data. The rest 90% is for training, of which 10% are split as validation data. We use validation data for model selection. The final output model is selected with the highest validation accuracy.

**Noisy labels generation.** For clean datasets, we artificially corrupt the class labels of training and validation sets following the instance-dependent label noise generalization method in Xia et al. (2020). We generate noisy datasets of {0.1, 0.2, 0.3, 0.4} four noise levels.

**Network structure and optimization.** For a fair comparison, all experiments are conducted on NVIDIA GeForce RTX 2080 Ti, and all methods are implemented by PyTorch. The dimension of background variable $U$ is set to 2. We employ a three-layer MLP with the Softsign activation function for every single model. The batch size is set to 128. We use SGD optimizer with momentum 0.9 and an initial learning rate 0.001. Learning rate is updated by *ReduceLROnPlateau*, which reduces learning rate when a metric (here we choose training loss as the metric) has stopped improving.

**Baselines.** We compare our methods R-$p$-Fair and RCFC with six baselines of four types:
- Standard supervised learning (SSL). It takes all the features as input and noisy labels as the target, which is not fair.
- $p$-Fair (Zafar et al., 2017). It takes all the features except $A$ as input and noisy labels as the target, which is softly fair with fairness metric $p$ value. We reimplement this method with Pytorch.
- Ablation-U (Ab-U). It postulates background variable $U$ but does not model the label noise.
- Ablation-N (Ab-N). It models the label noise but does not postulate background variable $U$.
- Counterfactual fairness learning (CFL) (Kusner et al., 2017) . It only uses non-descendants to make predictions, which is counterfactually fair.
- Counterfactual fairness learning with Mixup (CFL-M). Based on CFL, it additionally applies mixup technique, which is both counterfactually fair and kind of robust to label noise.

**Results.** The results in Table 2 and Table 4 show that there is a steady lift of our method on the classification accuracy and fairness score, compared with baseline methods. However, all statistical methods suffer from label noise to a great extent. It is because IDLN is ill-defined and handling it with purely statistical relations is not sufficient.

Table 3: Means and Standard deviations of classification accuracy on ADULT dataset over 5 trials.

| ADULT | 0.1 | 0.2 | 0.3 | 0.4 |
|---|---|---|---|---|
| Ab-U | 65.13±1.97 | 65.16±2.39 | 64.10±3.98 | 61.03±9.97 |
| Ab-N | 73.07±2.46 | 73.30±2.08 | 70.61±2.44 | 61.41±9.27 |
| CFL | 66.17±1.22 | 67.20±3.24 | 63.27±5.67 | 61.11±8.69 |
| CFL-M | 69.07±2.86 | 64.65±3.79 | 64.42±4.04 | 63.90±4.29 |
| RCFC | **74.69±0.42** | **74.65±0.42** | **74.30±0.44** | **72.96±1.91** |

Table 4: Means and Standard deviations of classification accuracy and fairness score ($p$ value) on BANK dataset over 5 trials.

| BANK | IDLN-0.1 | | IDLN-0.2 | | IDLN-0.3 | | IDLN-0.4 | |
|---|---|---|---|---|---|---|---|---|
| | Accuracy | Fairness | Accuracy | Fairness | Accuracy | Fairness | Accuracy | Fairness |
| SSL | 84.54±4.23 | 35.49±15.27 | 66.60±12.87 | 19.49±5.63 | 58.69±4.92 | 18.51±5.95 | 56.31±4.90 | 17.80±5.64 |
| $p$-Fair | 87.22±1.80 | 43.08±21.69 | 67.00±13.00 | 18.65±5.34 | 58.62±5.04 | 18.53±5.69 | 57.16±5.20 | 17.90±5.54 |
| R-$p$-Fair | 87.18±2.33 | 49.57±27.26 | 76.02±7.08 | 25.20±7.08 | 60.43±4.90 | 18.54±4.88 | 60.71±6.36 | 19.32±5.02 |

Table 5: Means and Standard deviations of classification accuracy on BANK dataset over 5 trials.

| BANK | 0.1 | 0.2 | 0.3 | 0.4 |
|---|---|---|---|---|
| Ab-U | 87.19±1.78 | 85.86±2.19 | 81.48±4.93 | 68.43±11.35 |
| Ab-N | 87.51±0.71 | 86.75±1.37 | 85.52±4.08 | 76.74±9.81 |
| CFL | 84.09±7.23 | 72.09±14.92 | 58.29±16.08 | 55.42±14.11 |
| CFL-M | 80.40±7.70 | 74.09±12.35 | 64.98±8.41 | 58.85±8.32 |
| RCFC | **88.77±0.61** | **88.76±0.62** | **88.72±0.61** | **86.40±1.88** |

The results in Table 3 and Table 5 demonstrate that our method achieves distinguished classification accuracy and is counterfactually fair. Compared with counterfactually fair methods CFL-M and CFL, our method additionally extracts as much as possible knowledge from the data in the precondition of satisfying a fairness requirement. These credits should go to the postulated causal graph which captures the data structure well. The results of ablation studies Ab-U and Ab-N show that exploiting causal relations and modeling label noise are both significant.

As the noise rate increases, the accuracy of all baselines decreases significantly while there is just a slight drop for our method. Even for challenging noise rates of 0.4, our method achieves good accuracy, uplifting about 15 and 30 points on ADULT and BANK, respectively. CFL-M and CFL have similar performances on both datasets, which means the mixup technique itself does not handle the instance-dependent label noise effectively. It also reflects the improvements of our method are mainly benefited from the proposed causal model which contributes to the identifiability of the transition relationship between clean and noisy labels.

## 6. Conclusions

This paper proposed general frameworks for learning fair classifiers with instance-dependent label noise. We notice that label noise not only degenerates the classification accuracy but misleads the fairness-aware algorithms even more prejudiced than fairness-unaware ones. We adapt statistically fair methods to the label noise setting and build consistent classifiers. Then we postulate a general causal graph, which can interpret the real-world datasets well. By exploiting the causal graph, we design an algorithm that both strictly achieves counterfactual fairness and identifies the transition relationship between clean and noisy labels. Experiments conducted on benchmark datasets demonstrate the effectiveness of our method.

## Acknowledgments

SW was partially supported by Australian Research Council Projects DE-190101473. MG is supported by ARC DE210101624. BH was supported by the RGC Early Career Scheme No. 22200720 and NSFC Young Scientists Fund No. 62006202. YL is partially supported by the NSF FAI program in collaboration with Amazon under grant IIS-2040800. TL was partially supported by Australian Research Council Projects DE-190101473, IC-190100031, and DP-220102121. We thank anonymous reviewers for their constructive comments.

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

## Appendix A.  Derivation process of getting $T_y(a)$ from $T_y(x)$

Here we take $T_{-1}(a)$ as an example. Let $X' \triangleq (Z, W)$ and $X_a \triangleq (A = a, Z, W)$:

$$
\begin{aligned}
T_{-1}(a) &= P(\tilde{Y} = +1 \mid Y = -1, A = a) \\
&= \frac{\int P(X' = x', \tilde{Y} = +1, Y = -1, A = a)\,\mathrm{d}x'}{\int P(X' = x', Y = -1, A = a)\,\mathrm{d}x'} \\
&= \frac{\int P(\tilde{Y} = +1 \mid X = x_a, Y = -1)P(Y = -1 \mid X = x_a)P(x_a)\,\mathrm{d}x_a}{\int P(Y = -1 \mid X = x_a,)P(x_a)\,\mathrm{d}x_a} \quad (16) \\
&= \frac{\int T_{-1}(x_a)P(Y = -1 \mid X = x_a)P(x_a)\,\mathrm{d}x_a}{\int P(Y = -1 \mid X = x_a)P(x_a)\,\mathrm{d}x_a},
\end{aligned}
$$

where $P(Y = -1 \mid X = x) = \frac{P(\tilde{Y} = -1 \mid X = x) - T_{+1}(x)}{(1 - T_{-1}(x) - T_{+1}(x))}$. In practice, we can use the above equation to consistently estimate $T_{-1}(a)$.

## Appendix B.  Derivation process of the negative ELBO

To derive the ELBO, we start with maximizing the data likelihood $p_\Theta(A, Z, W, \tilde{Y})$:

$$
\log p_\Theta(a, z, w, \tilde{y}) \tag{17}
$$

$$
= \log \int_u \int_y p_\Theta(a, z, w, \tilde{y}, y, u)\mathrm{d}y\,\mathrm{d}u \tag{18}
$$

$$
= \log \int_u \int_y \frac{p_\Theta(a, z, w, \tilde{y}, y, u)}{q_{\phi_1}(u \mid z)q_{\phi_2}(y \mid u, z)}q_{\phi_1}(u \mid z)q_{\phi_2}(y \mid u, z)\mathrm{d}y\,\mathrm{d}u \tag{19}
$$

$$
= \log \mathrm{E}_{(u,y)\sim q_\phi(u,y\mid z)}\left[\frac{p_\Theta(a, z, w, \tilde{y}, y, u)}{q_{\phi_1}(u \mid z)q_{\phi_2}(y \mid u, z)}\right] \tag{20}
$$

$$
\geq \mathrm{E}_{(u,y)\sim q_\phi(u,y\mid z)}\left[\log \frac{p_\Theta(a, z, w, \tilde{y}, y, u)}{q_{\phi_1}(u \mid z)q_{\phi_2}(y \mid u, z)}\right] \triangleq \text{ELBO} \qquad \text{(Jensen's Inequality)} \tag{21}
$$

$$
= \mathrm{E}_{(u,y)\sim q_\phi(u,y\mid z)}\left[\log \frac{p(a)p(u)p_{\theta_1}(z \mid u)p_{\theta_2}(y \mid u, z)p_{\theta_3}(w \mid u, a, z, y)p_{\theta_4}(\tilde{y} \mid a, z, y, w)}{q_{\phi_1}(u \mid z)q_{\phi_2}(y \mid u, z)}\right] \tag{22}
$$

$$
= \mathrm{E}_{(u,y)\sim q_\phi(u,y\mid z)}\left[\log p_{\theta_1}(z \mid u)\right] + \mathrm{E}_{(u,y)\sim q_\phi(u,y\mid z)}\left[\log p_{\theta_3}(w \mid u, a, z, y)\right] \tag{23}
$$

$$
+ \mathrm{E}_{(u,y)\sim q_\phi(u,y\mid z)}\left[\log p_{\theta_4}(\tilde{y} \mid a, z, y, w)\right] \tag{24}
$$

$$
+ \mathrm{E}_{(u,y)\sim q_\phi(u,y\mid z)}\left[\log \frac{p(u)}{q_{\phi_1}(u \mid z)}\right] + \mathrm{E}_{(u,y)\sim q_\phi(u,y\mid z)}\left[\log p(a)\right] \tag{25}
$$

$$
= \mathrm{E}_{(u,y)\sim q_\phi(u,y\mid z)}\left[\log p_{\theta_1}(z \mid u)\right] + \mathrm{E}_{(u,y)\sim q_\phi(u,y\mid z)}\left[\log p_{\theta_3}(w \mid u, a, z, y)\right] \tag{26}
$$

$$
+ \mathrm{E}_{(u,y)\sim q_\phi(u,y\mid z)}\left[\log p_{\theta_4}(\tilde{y} \mid a, z, y, w)\right] - \mathrm{D}_{\mathrm{KL}}(q_{\phi_1}(u \mid z)\|p(u)) + \log p(a). \tag{27}
$$

Since $\log p(a)$ is a constant, we drop it in ELBO. Then, the final negative ELBO can be defined as:

$$
-\text{ELBO} \triangleq -\mathrm{E}_{(u,y)\sim q_\phi(u,y\mid z)}\left[\log p_{\theta_1}(z \mid u)\right] - \mathrm{E}_{(u,y)\sim q_\phi(u,y\mid z)}\left[\log p_{\theta_3}(w \mid u, a, z, y)\right] \tag{28}
$$

$$
- \mathrm{E}_{(u,y)\sim q_\phi(u,y\mid z)}\left[\log p_{\theta_4}(\tilde{y} \mid a, z, y, w)\right] + \mathrm{D}_{\mathrm{KL}}(q_{\phi_1}(u \mid z)\|p(u)). \tag{29}
$$

