# OpenReview forum: "Fair Classification with Instance-dependent Label Noise"
_cclear.cc/CLeaR/2022/Conference — CLeaR 2022 Poster_

### Official Review · Reviewer_2xWR · 2021-11-21

**Confidence:** 4
**Overall Score:** 6

**Main Review:**

Originality: The paper considered an interesting situation in which there are label noises on the outcome which renders most state of the art fairness-aware methods collapsing. This is a new problem to consider in the fairness literature and the authors provided an initial attempt to solve the issue.

Significance: The paper addressed an issue in the fairness background which is clearly relevant to the CLeaR community. The method did not significantly advance the state-of-the-art but rather I feel it is an immature attempt to solve the problem, but the results in the paper indeed provided new insights to the research problem.

Technical quality: The proposed approach appears reasonable and has been substantiated by empirical results but not theoretical results.

Clarity: The draft is badly written, at least in the current forms, especially the experiments section. Table 2 and table 4 have exactly the same caption, which did not provide enough information for the readers. I also spent some time finding where the R-p-fair is defined. I believe the draft needs rewriting, otherwise, as a reader, I have to take a lot of guesses.

**Summary:**

The paper considered an important machine learning direction, the fairness issue, in the context when there is labeling noise in the outcome. The paper showed the existing method may suffer from the existence of certain types of label noises and therefore the authors developed robust method for both statistical fairness and counterfactual fairness to account for those. The proposed methods are considered together with other state-of-the-art in two real datasets.

---

> ### Author Response · Authors · 2021-12-04
> **Response to Reviewer 2xWR**
>
> Thank you for your time and efforts in reviewing our paper. We would like to address your concerns as follows.
>
> Originality: Thank you for your recognition of our originality.
>
> Significance: Our method consistently surpasses all the baselines significantly. We are not sure which state-of-the-art method that our method did not significantly advance. Could you name one and we would love to conduct additional experiments for it?
>
> Technical quality: There are mainly two parts of the theoretical analysis of our paper. First, we theoretically analyzed how to build robust and fair classifiers for statistical fairness notions. Second, the classifier we build in Section 4 is theoretically guaranteed to be counterfactually fair.
>
> Clarity: Thanks for your constructive suggestions. We will revise them in our final version.

---

### Official Review · Reviewer_uZWX · 2021-11-22

**Confidence:** 3
**Overall Score:** 6

**Main Review:**

Pros:
1. This paper is well motivated. Learning a fair classifier with instance-dependent label noise is a novel, interesting and significant problem.
2. This paper is presented clearly and organized logically.
3. This paper proposes a general framework and considers fairness including both statistical fairness notions and causality-based fairness notions. In addition, the methods are well supported by theoretic analysis.
4. Extensive experimental results are reported on two real datasets and compared with six representative methods.

Cons:
1. My main concern is the rationality of the constructed causal graph. Although the authors take ADULT dataset as an example to demonstrate the proposed causal graph interprets the data, it is uncertain whether the construction of the causal graph is complete. For example, what happens when the parents of A exist?
2. Table 1 shows that the label noise makes fairness-aware algorithms even more prejudiced than fairness-unaware ones, and the instance-dependent label noise is a more challenging setting compared with class-dependent label noise. It would be better if the authors can provide a deeper analysis of this phenomenon.
3. It would be better to add some discussion about the relationship and distinction between this paper and related work.


**Summary:**

This paper proposes a novel problem that learning a fair classifier with instance-dependent label noise. The authors provide a general framework to deal with this problem for both statistical fairness notions and causality-based fairness notions. Experimental results show the effectiveness of the proposed methods.

---

> ### Author Response · Authors · 2021-12-04
> **Response to Reviewer uZWX**
>
> Thank you for your time and efforts in reviewing our paper. We would like to address your concerns as follows.
>
> ## The completeness of causal graph
> Specifically, in our method, A is only used to reconstruct W and $\tilde{Y}$, i.e., $P(W \mid A \cdots)$, $P(\tilde{Y} \mid A \cdots)$. When the parents of A exist, conditioned on A, W and $\tilde{Y}$ (the descendants of A) are independent on the parents of A. Therefore, the existence of parents of A does not affect our method.
>
> Generally. since the true causal graph of a specific dataset is always agnostic, we can only postulate or approximately discover the causal graph, both of which cannot guarantee the completeness of the causal graph.
>
>
> ## Analysis of the effects of label noise on fairness-aware algorithms
> Thanks for your valuable feedback. For UC, the method which optimizes the training loss unconstrainedly, the classifier is misled to fit the noisy data. For fairness-aware algorithms with general formulation as follows:
> \begin{equation}
> \begin{aligned}
>     &\text{minimize} \quad \sum_{n=1}^{N}L(f(x_n), y_n) \\\\
>     &\text{subject to} \quad \text{Fair}(X, Y, f) = 0.
> \end{aligned}
> \end{equation}
> The classifier is misled by both classification loss and wrong fair metric. With different fair metrics, the classifier is misled to different degrees. Therefore, the performance of fairness-aware algorithms has no guarantee and could be worse than fairness-unaware ones.
>
>
> ## Related works
> Thanks for your constructive suggestion! As Reviewer 2xWR agreed, this is a new problem. Thus, to the best of my knowledge, there is no work that is related enough to be discussed and distinguished. We would definitely appreciate it if you could bring up some similar works.

---

### Official Review · Reviewer_x5wb · 2021-11-24

**Confidence:** 3
**Overall Score:** 5

**Main Review:**

The paper proposes an analysis of learning with fairness constraints in the presence of label noise. The difficulty of the framework is that the authors study the general problem of instance-dependent noise, while previous work focused on simpler notions of noise such as class-conditional noise.

The authors first study the setup of loss minimization under equalized odds constraints. essentially assuming that the noise model is known. Then the authors study a special case of counterfactual fairness (based on the model in Figure 1 and instanciated on the Adult dataset), and propose a variant of the algorithm of Kusner et al. to perform counterfactually fair prediction when the ``true'' label is not observed.

Overall the paper is well written, even though the assumptions are not entirely clear (see below). The novelty and originality is limited because there is a wide literature on both fair ML with label noise and counterfactual fairness, and the paper stays close to existing approaches. Also, a large part of the paper is devoted to calculations that could be deferred to an Appendix. Experiments are carried out thoroughly with relevant baselines.

pros: compared to the counterfactual fairness paper (by Kusner et al.), I like the idea of distinguishing the observed biased label, a ground truth label Y and a counterfactually fair label that is learnt. This allows to avoid a problem of the original paper that it was impossible to evaluate whether the algorithm was close to an underlying truth or not, since it was not clear what the "ground truth" was.

some comments:
* The value of Section 3 to the paper is unclear to me, since my understanding was that the intention was to treat the case where the noise model is unknown.
* It would be better to spell out from the start of section 3 that the assumption is that the transition matrices are unknown
* In Eq 7 I don't understand what T_{-1} is. The dependency of T on X seems to have magically disappeared, and I do not see how that is possible without making additional assumptions (e.g., that the noise only depends on the protected group, but not on arbitrary features of X
* The model used in the section on counterfactual fairness is clear but not clearly motivated. Since U only depends on Z, I don't understand the value of having functions of U for W and Y -> why not simply functions of Z directly?
* the motivation underlying mixup in that setting is not clear to me, in particular it is introduced when "f is miled by tilde{Y}" (what does "mislead means") and then the justification is that "mixup increases the sample complexity of the training set" (what is the sample complexity of a dataset?)


**Summary:**

review

---

> ### Author Response · Authors · 2021-12-04
> **Response to Reviewer x5wb (1)**
>
> Thank you for your time and efforts in reviewing our paper. We would like to address your concerns as follows.
>
>
> ## The value of Section 3
>
> Please allow me to first elaborate on the logical structure of this paper.
> 1. First, taking fairness and label noise issues into consideration is significant in the era of big data.
> 2. Second, we notice that noisy labels, unfortunately, make original fairness-aware algorithms even more prejudiced than fairness-unaware ones, and thereby harmful.
> 3. Fairness notions can be generally divided into statistical fairness notions and causality-based fairness notions.
> * For statistical fairness notions, since they share a general formulation, we can provide a general solution to them, i.e., we rewrite the classification risk and the fairness metric in terms of noisy data and thereby build robust classifiers.
> * For causality-based fairness notions, the formulations vary a lot, so we only show how to design a robust and counterfactually fair classifier.
> 4. At last, we conclude our paper.
>
> Overall, our work intended to provide general frameworks for learning (statistically or causally) fair classifiers with instance-dependent label noise. Therefore, **Section 3 plays an important role to show how to design robust classifiers with statistical fairness notions.**
>
>
> ## Transition matrix
> We didn’t assume that the transition matrices are known or unknown. The transition matrix is a theoretical concept that builds the bridge between clean class posterior and noisy class posterior. In the main part of Section 3, we just **theoretically analyzed** how to build robust classifiers with it. As for how to **acquire it**, we stated that in practical implementation, we can estimate it if it is not given at the end of Section 3.
>
> ## Dropping $x$ in $T$
> Thanks for your valuable feedback about this point. Note that equalized odds is a group fairness notion and the term in Eq. (7) we want to rewrite does not involve $x$. Therefore, we have to consider the transition relations at group level. Basically, there are two ways to achieve this:
> 1. We can straightforwardly take the integral of $ P(X=x, \hat{Y}=1 \mid A=a, Y=1) $, which involves $x$, over x:
> \begin{equation}
>     \begin{aligned}
> &P(\hat{Y}=1 \mid A=a, Y=1)\\\\
> &= \int P(X=x, \hat{Y}=1 \mid A=a, Y=1) \mathrm{~d} x \\\\
> &= \int \frac{P(X=x, \hat{Y}=1, A=a, Y=1)}{P(A=a, Y=1)}  d x \\\\
> &= \int \frac{P(X=x, \hat{Y}=1, A=a, Y=1)P(X=x, A=a, \hat{Y}=1)}{P(A=a, Y=1)P(X=x, A=a, \hat{Y}=1)} d x \\\\
> &= \int \frac{P(Y=1 \mid X=x, A=a, \hat{Y}=1)P(X=x, A=a, \hat{Y}=1)}{P(A=a, Y=1)} d x \\\\
> &= \int \frac{\left(P(\tilde{Y}=1 \mid X=x, A=a, \hat{Y}=1) - T_{-1}(x)\right) P(X=x, A=a, \hat{Y}=1)}{\left(1-T_{-1}(x)-T_{+1}(x)\right) P(A=a, Y=1)}d x,
>     \end{aligned}
> \end{equation}
> $ P(A=a, Y=1) $ can be further rewritten as:
> \begin{equation}
>     \begin{aligned}
>     &P(A=a, Y=1)\\\\
>     &= \int P(X=x, A=a, Y=1) d x \\\\
>     &= \int P(Y=1 \mid X=x, A=a) P(X=x, A=a) d x \\\\
>     &= \int \frac{\left(P(\tilde{Y}=1 \mid X=x, A=a) - T_{-1}(x)\right) P(X=x, A=a)}{\left(1-T_{-1}(x)-T_{+1}(x)\right)} d x
>     \end{aligned}
> \end{equation}
> The first way is a little bit complex, and thus we consider the average transition relations
>
> 2. We define the average transition relations as $ T_{-1} = P(\tilde{Y}=+1 \mid Y=-1)$. Then we have:
> \begin{equation}
>     \begin{aligned}
>     T_{-1} &=P(\tilde{Y}=+1 \mid Y=-1)\\\\
>     % &= \frac{P(\tilde{Y}=+1, Y=-1)}{P(Y=-1)} \\\\
>     &= \frac{\int P(X=x, \tilde{Y}=+1, Y=-1)d x}{\int P(X=x, Y=-1)d x} \\\\
>     &= \frac{\int P(\tilde{Y}=+1 \mid X=x, Y=-1) P(Y=-1 \mid X=x) P(x) d x}{\int P(Y=-1 \mid X=x) P(x) d x} \\\\
>     &= \frac{\int T_{-1}(x) P(Y=-1 \mid X=x) P(x) d x}{\int P(Y=-1 \mid X=x) P(x) d x}, \\\\
>     \end{aligned}
> \end{equation}
>     where  $P(Y=-1 \mid X=x) = \frac{P(\tilde{Y}=y \mid X=x) - T_{-y}(x)}{\left(1-T_{-1}(x)-T_{+1}(x)\right) P(\tilde{Y}=y \mid X=x)}$
>
> Thus, we do not need additional assumptions to drop $X$ in $T$. We are sorry we did not make it clear in the paper and we will elaborate it in our final version. Thanks again for your constructive suggestion!

---

> ### Author Response · Authors · 2021-12-04
> **Response to Reviewer x5wb (2)**
>
> ## The model used in the section on counterfactual fairness
>
> According to the causal graph, U depends on Z, W, and Y.
> In the decoding phase, since U is a parent of Z, W, and Y, U should be in the functions of Z, W, and Y of the causal decomposition Eq. (10).
> In the encoding phase, although U depends on Z, W, and Y, we infer U only from Z because (1) Y is unknown; (2) W is the descendant of A, which cannot be used in the decision function.
>
> ## Mixup
> The problem here is that the counterfactually fair prediction Y is a latent intermediate variable, which means the map between Y (cluster id) and the semantic class label is lost. To map the prediction Y to the semantic class label, and also keep prediction Y from overfitting to noisy label (the meaning of ‘misled’), we employ Mixup to learn this map.
>
> We misused the *sample complexity*. We will change it to *complexity*. Thanks for your valuable feedback.

---

### Decision · Program_Chairs · 2022-01-12

**Decision:**

Accept (Poster)

**Comment:**

The reviewers generally agreed that this paper makes a contribution to fair learning. There were some questions and confusions that were largely ameliorated in the authors' response. (One reviewer did not respond to the response, but I read both the review and response and felt that the authors cleared up the confusion.) This paper is above the threshold for acceptance.